# Finite Elements Analysis of Tooth—A Comparative Analysis of Multiple Failure Criteria

**DOI:** 10.3390/ijerph20054133

**Published:** 2023-02-25

**Authors:** Radu Andrei Moga, Cristian Doru Olteanu, Botez Mircea Daniel, Stefan Marius Buru

**Affiliations:** 1Department of Cariology, Endodontics and Oral Pathology, School of Dental Medicine, University of Medicine and Pharmacy Iuliu Hatieganu, Str. Motilor 33, 400001 Cluj-Napoca, Romania; 2Department of Orthodontics, School of Dental Medicine, University of Medicine and Pharmacy Iuliu Hatieganu, Str. Avram Iancu 31, 400083 Cluj-Napoca, Romania; 3Department of Structural Mechanics, School of Civil Engineering, Technical University of Cluj-Napoca, Str. Memorandumului 28, 400114 Cluj-Napoca, Romania

**Keywords:** tooth, enamel, dentin, stress absorption–dissipation ability, periodontal breakdown, Finite elements analysis (FEA), failure criteria selection, orthodontic movements

## Abstract

Herein Finite elements analysis (FEA) study assesses the adequacy and accuracy of five failure criteria (Von Mises (VM), Tresca, maximum principal (S1), minimum principal (S3), and Hydrostatic pressure) for the study of tooth as a structure (made of enamel, dentin, and cement), along with its stress absorption–dissipation ability. Eighty-one 3D models of the second lower premolar (with intact and 1–8 mm reduced periodontium) were subjected to five orthodontic forces (intrusion, extrusion, tipping, rotation, and translation) of 0.5 N (approx. 50 gf) (in a total of 405 FEA simulations). Only the Tresca and VM criteria showed biomechanically correct stress display during the 0–8 mm periodontal breakdown simulation, while the other three showed various unusual biomechanical stress display. All five failure criteria displayed comparable quantitative stress results (with Tresca and VM producing the highest of all), showing the rotational and translational movements to produce the highest amount of stress, while intrusion and extrusion, the lowest. The tooth structure absorbed and dissipated most of the stress produced by the orthodontic loads (from a total of 0.5 N/50 gf only 0.125 N/12.5 gf reached PDL and 0.01 N/1 gf the pulp and NVB). The Tresca criterion seems to be more accurate than Von Mises for the study of tooth as structure.

## 1. Introduction

During the orthodontic treatment, the tooth structures are the first tissues subjected to the orthodontic forces (i.e., through bracket) or/and to other types of loads. Some of the stresses produced by the functional/parafunctional/orthodontic forces are dissipated by and through the tooth components (each one with a different micro-architecture), while other stresses are further transmitted to the periodontal ligament (PDL), dental pulp, and the surrounding bone (trabecular and cortical) [1,2,3,4]. When analyzing the biomechanical behavior of tooth structures under loads (i.e., absorption and dissipation of stresses by the structures), understanding the process involves the knowledge of the inner anatomical micro-architecture of each component (i.e., enamel, cement, and dentin) [1,4].

Tooth structures are highly mineralized tissues with physical properties based on composition and micro-morphology [5]. The bracket is usually made from stainless steel (a ductile material) and cemented to the enamel.

Enamel is considered to be the hardest tissue in the human body (92–96% inorganic matter, mostly hydroxyapatite; 1–2% organic material; 3–4% water) [5]. The micro-architecture of enamel is of structural units containing hydroxyapatite (crystalline calcium phosphate) organized and arranged in the form of hexagonal-prism-shaped rods of 0.003–0.005 mm [1,5]. From the biomechanical point of view, enamel has a high mineral content (i.e., anisotropy), and is considered to be resembling more a brittle material (high elastic modulus, low tensile strength, and high compressive strength) with predisposition to cracks and fracture (depending on the direction of the applied stress and prismatic orientations) [1,5]. The enamel thickness varies over the tooth surface, thicker at the cusps and thinner to the proximal sides, being reported to be higher distal (1.21–1.8 mm, lower premolars) than mesial (1.15–1.42 mm, second lower premolars) [1,6].

Inside the enamel component is the dentin–enamel junction (DEJ), seen as an interface with a role in dissipating stress (i.e., ductile resemblance) [5]. The DEJ covers the dentin component, which is less mineralized than enamel (i.e., 70% hydroxyapatite, 18–20% organic material, and 10–12% water), with a micro-architecture of phosphoric apatite crystallites. The dentin ensures the absorption and dissipation of the bite/orthodontic force and has physical properties and structural components that are varying depending on topography (resembling more a ductile material but with a certain brittle component) [1,5,7]. The structural composition consists of oriented tubules surrounded by a highly mineralized cuff of peritubular dentin and an inter-tubular matrix of Type I collagen fibrils reinforced with hydroxyapatite [5,7]. The micro-architecture of dentin (i.e., tubules, peritubular dentin, collagen of inter-tubular dentin, and lower mineral content) ensures a lower modulus of elasticity and micro-hardness (low elastic modulus, high tensile strength, and high compressive strength), varying profoundly depending on location, and with significant effects over the force of absorption–distribution in the tooth structures [5,7]. The dentin component represents most of the structural volume of the tooth.

Beneath the dentin is the pulp chamber containing the dental pulp (which resembles a ductile material) [2]. Around the pulp chamber, there is a layer of cells (odontoblast) involved in dentin formation connected with dentinal tubules by channels [1].

Cementum is another mineralized component of the tooth structures, covering the entire root (i.e., the dentin component), ensuring support and load absorption–dissipation, and resembling a ductile material but with a certain degree of brittle flow [8]. Thus, from all the tooth structures, the dentin and cementum (i.e., representing almost the entire tooth structural volume) ensure the absorption, reduction, and dissipation of applied forces (occlusal loads: functional/parafunctional/orthodontic), enabling the entire tooth structure to behave biomechanically as a ductile material but with a certain brittle flow mode [1,2,5,8].

The components of the tooth structure are anisotropic and non-homogenous, with variable properties depending on the experimental circumstances (i.e., shapes, dimensions, and environment), and do not follow Hook’s law [1]. Thus, for a better understanding of the biomechanical behavior of the tooth structure under loads, the entire structures stress and strain quantitative and qualitative displays should be linked with the inner micro-architecture and must be validated through correlations with in vivo data and in vitro reports having similar experimental conditions [1,2,3,5].

There are numerous in vivo, in vitro, and Finite elements analysis (FEA) studies investigating the fracture resistance of the tooth and its components, along with the biomechanical behavior under functional and/or orthodontic loads. In vitro studies usually assessed the fracture resistance (ultimate tensile-compressive strengths) of intact, endodontically treated or directly restored with different composite material teeth [1,5,9]. Nevertheless, an extracted tooth subjected to in vitro tests suffers the variable change of its physical properties [1]. A recent study reported serious discrepancies between the design of an FEA study and in vitro validation related to the supporting tissues modeling, with the FEA considered to be the better method of investigation [10]. FEA studies accurately reconstruct the anatomical PDL and bone, opposingly to the in vitro studies where the tooth is embedded in varied materials (i.e., epoxy resin, composite resin, or silicone), which cannot mimic the anatomical tissues [10]. There are no reports regarding neither the clinical functional correlation between tooth–PDL–dental pulp during their biomechanical behavior, nor the adequate FEA method (based on these relationships) to be employed [2,3].

Each method of studying the tooth and its surrounding periodontium suffers of various degrees of limitation [1]. In vivo and in vitro studies cannot supply an individualized detailed quantitative and qualitative stress distribution for each component [2,3]. Only FEA studies can individually assess each component’s behavior but cannot accurately simulate neither the clinical situation and conditions, nor the micro-architecture of the tissues, thus needing validation through correlations with in vivo and in vitro data [2,3].

There are numerous quantitative in vitro reports regarding the physical properties of each tissue of the tooth (extracted specimens tested under various loads) that can be used as indirect FEA validation criteria: maximum compressive stress—62.2 MPa enamel, 193.7 MPa dentin, and 126.1 MPa enamel–dentin (premolars and canines) [1]; maximum tensile strength—11.5–42.1 MPa enamel, 33.9–61.6 MPa dentin, 46.9 MPa enamel–dentin (third molar) [5]; maximum shear stress—29–73.1 MPa dentin, 53.9–104 MPa enamel–dentin (third maxillary molar) [7]; hardness—630–730 MPa dentin, 370–460 MPa cementum (lower molars) [9]. In vivo dental studies reported that if the maximum hydrostatic pressure (MHP) of 16 KPa (another validation criterion) exceeds ischemia, necrosis, resorption, and further tissue loss shortly follows.

FEA engineering simulations are accurate due to the employment of the adequate failure criteria based on the type of the analyzed material (i.e., the theory of yielding of materials) and the correct input data. The yielding theory is based on the several types of stress deformation displayed by the materials when subjected to a force before the fracture or destruction occurs [2,3]. There is a wide engineering knowledge related to the type of material (brittle, ductile, liquid or gas) and the specific failure criteria to be employed, which is practically unstudied in the dental field [2,3,11].

FEA dental analysis is regarded with care due to various qualitative and quantitative reports frequently contradicting in vivo and in vitro data, as a consequence of the misunderstanding of the failure criterion principles (e.g., yielding theory) [2,3]. There are numerous studies employing various failure criteria: Von Mises (VM) overall stress [10,11,12,13,14,15,16], Tresca (shear stress), maximum principal (S1-tensile stress) [11,13,17,18,19,20], minimum principal (S3-compressive stress) [11,17,18,20,21,22,23], and pressure (hydrostatic pressure) [24,25,26,27,28], without addressing the correlations between the tissues’ inner anatomical micro-structure, the material-type resemblance (ductile vs. brittle), the suitability of the criteria, the biomechanical behavior of the tooth correlated with that of PDL and with the maximum hydrostatic pressure (MHP), the force dissipation and absorption, and the biomechanically correct stress display (all seen as mandatory validation criteria). Moreover, no justification regarding the selection and employment of one failure criterion vs. another for the living dental tissues was found, except in our previous studies [2,3]. An earlier study examined the adequacy of failure criteria only for the endodontic restoration materials without any reference to the dental tissues [11].

It must be emphasized that FEA is a mathematical algorithm-based method employing failure criteria specially design for a certain type of biomechanical behavior and offering correct results (both qualitative and quantitative) only for that material type. The ductile materials (e.g., rubber and streel) suffer various degrees of elastic or plastic deformation, while the brittle materials (e.g., ceramic, glass, stone, and concrete) suffer no or limited plastic deformation before fracture [2,3,11].

VM and Tresca were designed for ductile materials, S1 and S3 for brittle, while pressure/hydrostatic pressure for liquid/gas. Limited knowledge regarding the resemblance of living dental tissues to brittle or ductile material categories is available [2,3,10,24,25,26,27,28,29,30].

Nonetheless, the proper selection of the failure criteria is mandatory for the correctness of the results and conclusions, as well as the anatomical correctness of the analyzed tissue model. Two recent studies of our group [2,3] reported the PDL and dental pulp, with its neuro-vascular bundle (NVB), resembles more ductile materials but with a certain brittle flow mode, and Von Mises and Tresca are adequate (with Tresca seen as more adequate) for their study. For the tooth structure (i.e., enamel, dentin, and cement), no similar studies have been found (despite their need).

This analysis assessed the adequacy and accuracy of the five most used failure criteria for the study of the tooth as a structure (made of enamel, dentin, and cement), by simulating five orthodontic movements (intrusion, extrusion, rotation, translation, and tipping), under an applied force of 0.5 N (approx. 50 gf), over the second lower premolar, and in a gradual horizontal periodontal breakdown (0–8 mm). Additionally, the stress absorption–dissipation ability of the tooth as a structure was also examined.

## 2. Materials and Methods

This FEA analysis is part of a larger research project (clinical protocol 158/02.04.2018) incrementally developed, aiming to assess the biomechanical behavior of dental tissues (tooth and surrounding periodontium) under orthodontic loads in both intact and reduced periodontium [2,3]. The objective of the herein simulation is to identify the adequate failure criteria for the analysis of the tooth as a structure.

The present analysis examined eighty-one 3D models of the 2nd lower premolar and surrounding periodontium from nine patients (with a mean age of 29.81 ± 1.45 years, 4 males, 5 females, oral informed consent) in a total of 405 FEA simulations.

The inclusion criteria were: an intact and reduced non-inflamed periodontium, a satisfying oral hygiene status, a complete mandibular arch, intact lower premolars, no malposition, orthodontic treatment, regular follow-up availability. Considering that few patients met the inclusion criteria, and previous FEA studies used a sample size of only one and one or two tooth models, a sample size of nine conducted on eighty-one tooth models through a total of 405 simulations was considered adequate.

The base of our study was the result of CBCT (cone beam computed tomography) examination, using a 0.075 mm voxel size (ProMax 3DS, Planmeca, Helsinki, Finland) of the mandibular region containing the two premolars and molars.

Based on the CBCT data, a manual image reconstruction process (for enhanced anatomical accuracy) using Amira 5.4.0 software (Visage Imaging Inc. 300 Brickstone Square, Suite 201 Andover, MA, USA) was conducted by a single experienced clinician. The 2nd lower premolar (enamel, dentin, cement, and dental pulp) and surrounding periodontium (periodontal ligament, neuro-vascular bundle (NVB), and trabecular and cortical bones) components were identified and reconstructed based on Hounsfield shades of grey. The result was a total of nine 3D models (one for each patient) containing only the 2nd premolar and with various levels of bone loss (usually limited to the cervical third of the PDL), with 5.06–6.05 million C3D4 tetrahedral elements, 0.97–1.07 million nodes, and a global element size of 0.08–0.116 mm (extremely fine-grain mesh). The missing periodontium was reconstructed as accurately as possible, obtaining nine models with intact periodontium. The next step was to simulate a progressive horizontal bone resorption by reducing the bone and PDL by 1 mm in height (from 0 to 8 mm, Figure 1 and Figure 2) for each of the nine models, obtaining a total of eighty-one models. As a downside of the manual reconstruction technique, a limited number of surface anomalies/irregularities had been present in all models, but with quasi-continuity in all areas affected by stress, the internal processes (i.e., algorithm-based) have been passed. The mesh convergence testing was performed for all models. Both software (AMIRA and ABAQUS) prevent the mesh creation and the FEA analysis (i.e., due to internal algorithms) if many surface anomalies are present, allowing only a limited number of anomalies that do not interfere with the process. Nonetheless, a limited number of surface anomalies and irregularities was expected to remain even after several mesh-smoothing processes but without interfering with the analysis or results (e.g., the models presented in Figure 1 and Figure 2 displayed a total number of 264 element warnings (39 element warnings for the 665,501 elements of the tooth with bracket) from a total of 6.05 million elements, and no errors). All the element warnings were in areas where the stress concentrations are reduced, while areas with stress concentrations are quasi-continuous; thus, the accuracy of results was not altered.

The PDL was reconstructed with a variable thickness of 0.15–0.225 mm containing the neuro-vascular bundle of the 2nd premolar’s dental pulp. The cementum was reconstructed as dentin due to the similar physical properties (Table 1) and reduced volume percentage from the total amount of the tooth’s structural volume. The acknowledged physical properties were homogeneity, linear elasticity, perfectly bonded interfaces, and the base of the model with no displacements (i.e., similar with most of the other FEA studies of dental tissues).

The FEA analysis was performed in Abaqus 6.13–1 software (Dassault Systèmes Simulia Corp., Stationsplein 8-K, 6221 BT Maastricht, Netherlands), employing five failure criteria and simulating five orthodontic movements for each of the eighty-one models, reaching a total number of 405 simulations. The five failure criteria (currently used in the dental studies) were: Von Mises (overall stress), Tresca (shear stress), maximum principal (S1-tensile stress), minimum principal (S3-compressive stress), and pressure (hydrostatic pressure). The five simulated orthodontic movements were: intrusion, extrusion, tipping, rotation, and translation, under an orthodontic load of 0.5 N (approx. 50 gf) applied directly over the bracket (Figure 1 and Figure 2). The used amount of force was selected based on the fact that it is small enough to be safely applied in both the intact and reduced periodontium for all movements and for being able to correlate the results of this study with those previously reported for PDL, dental pulp, and NVB [2,3] (studied under similar conditions as herein).

The qualitative (i.e., color-coded projection of stress distribution areas on the tooth surface) and quantitative (i.e., the amount of stress) results were assessed and displayed in Figure 3, Figure 4, Figure 5, Figure 6 and Figure 7 and Table 2, Table 3 and Table 4. The quantitative results were correlated with in vitro reports regarding the physical properties of each tissue of the tooth [1,5,7,9] and with those previously reported for PDL, dental pulp, and NVB [2,3] (i.e., aiming to quantify how much of the tooth stress was absorbed and dissipated by the tooth structures and what quantitative amount reached the PDL and the dental pulp). The qualitative results were also correlated with other FEA reports [10,11,12,13,14,15,17,18,19,20,21,22,23,24,25,26,27,28], while their biomechanical correctness was assessed through a correlation with the acknowledged clinical mechanics and principles of the yielding theory of materials.

## 3. Results

This analysis involved a total number of 405 simulations on eighty-one 3D models of the second lower premolar from nine patients (nine models/patient). No noticeable influence of age, gender or periodontal status was visible. The results of simulations (both quantitative and qualitative) using all five failure criteria were displayed in Figure 3, Figure 4, Figure 5, Figure 6 and Figure 7 and Table 2 and Table 3.

Qualitatively, the stresses showed in the tooth’s structures were displayed as various color-coded projections in Figure 3, Figure 4, Figure 5, Figure 6 and Figure 7. Only Tresca and Von Mises showed almost similar color-coded stress projections for all five movements in both intact and reduced periodontium. The other three criteria showed various color-coded stress displays biomechanically unusual, sometimes resembling Tresca and VM displays, but no distribution pattern was visible during the entire periodontal breakdown simulation.

During the extrusion movement (Figure 3), in both the intact and reduced periodontium, Tresca (shear stress) and VM (overall stresses) displayed the biomechanically expected stress areas on both the vestibular and lingual side of the entire height of the root and around the bracket. Maximum principal (S1, tensile stress, positive sign values) displayed tensile stress only to the vestibular side of the root and around the bracket. Minimum principal (S3, compressive stress, negative sign values) showed the compressive stress areas only in the lingual side of the root and above the bracket. The pressure was limited to the vestibular side of the root and around the bracket. This stress distribution for S1 (vestibular tension), S3 (lingual compression) and pressure criteria (vestibular compression) on only one side of the root is biomechanically unusual (instead of the two sides of the tooth, which would be normal).

Throughout the periodontal breakdown simulation, the intrusion movement (Figure 4) showed the same pattern of color-coded stress area for the Tresca and VM as extrusion (i.e., lingual, and vestibular side of the root and around the bracket). The pressure criteria displayed vestibular tensions and lingual compression (unusual clear separation). The S3 criteria showed compression stress for the vestibular side and tension on the lingual side. A similar unusual stress display was visible for the S1 criteria (tension only lingually). The same biomechanically unusual behavior was visible for all of the above three criteria (i.e., the stress being displayed only to one side of the tooth instead of both sides).

As biomechanically expected, in the rotational movement (Figure 5), the VM and Tresca criteria displayed radicular stress in the cervical, middle, and apical third of the root (i.e., the distal-lingual and vestibular-mesial side). The other three criteria displayed visible stress areas on the bracket, while the color-coded display on the tooth (both coronal and radicular) was biomechanically inconclusive in tooth structures both in the intact and reduced periodontium.

When employing the Tresca and VM criteria for the tipping movement (0.5 N), the stress is displayed around the bracket and on the vestibular and lingual side of the root throughout its entire height (as expected), Figure 6. S1 tensile stress was localized on the vestibular-distal side of the root (positive sign stress, high tensile stress) and compression stress on the bracket. The S3 criteria displayed compressive stress areas around and on the bracket surface and the lingual side root stress (biomechanically: lingually there should be compression and vestibular tension). The pressure criteria displayed a stress area (positive sign) on the entire lingual side of the root (0–8 mm of loss) and negative sign stress areas on vestibular side (4–8 mm of loss), a biomechanical behavior where both compressive and tensile stresses are present, but with an unusual biomechanical display.

The same use of the Tresca and VM criteria in the translational movement (Figure 7) showed stress areas around and on the bracket surface, cervical third lingual side root stress for no bone loss, and cervical and middle-third root stress (i.e., vestibular, mesial, distal, and lingual) for the periodontal breakdown. For the entire periodontal loss simulation, the S3 criteria displayed a relatively correct compressive stress localized on the distal-vestibular root side (negative sign values), but almost no tensile component on the root mesial side). The S1 criteria displayed inconclusive areas of tensile stress (i.e., biomechanically incorrect) in the entire tooth structure for the intact periodontium. A reduced periodontium tensile stress was present on the mesial side, and a compressive component, on the distal-vestibular side. Pressure criteria showed various extents of the color-coded stress area (an inconclusive mix of positive and negative sign areas, biomechanically incorrect), on the distal and mesial side of root in reduced periodontium.

When using the Tresca and VM criteria, the stress at the bracket level was localized mostly around the bracket (and on the bracket surface for translation and rotation, movements that displayed the highest quantitative amount of stress). The other three criteria displayed various extents of the color-coded stress area around and on the bracket, with no visible pattern, most probably due to the different type of analyzed stress (i.e., tension, compression or hydrostatic pressure).

Quantitatively, the rotational and translational movements exhibited the highest amount of stress, while intrusion and extrusion, the lowest, for all five failure criteria (Table 2).

All five types of employed failure criteria displayed quantitative stress values in the intact and reduced periodontium (42.7–518 KPa (Tresca shear stress), 37.13–452.49 KPa (Von Mises overall stress), 45.61–325.42 KPa (max. Princ. S1 tensile stress), (−48.84)–(−446.96) KPa (min. Princ. S3 compressive stress), and 29.25–267.19 KPa (pressure hydrostatic stress)), significantly lower than the in vitro reported physical properties of tooth components (i.e., maximum compressive stress, maximum tensile strength, and maximum shear stress). The quantitative pressure results were the lowest of all five (i.e., nearly half if compared with Tresca/VM), while Tresca’s and Von Mises’ were the highest.

Solely based on the quantitative results (Table 2), a clear difference between the failure criteria (i.e., which is more adequate for the study of tooth structure) was not possible. However, the biomechanical analysis of the stress distribution (qualitative results and color-coded projections, Figure 3, Figure 4, Figure 5, Figure 6 and Figure 7), correlated with the known internal micro-structure involved in the absorption–dissipation stress, and the suitability of each failure criterion for a certain type of material (dentin and cement resembling ductile), showed Tresca and Von Mises to be more adequate than the other three criteria (with Tresca being more adequate for ductile non-homogenous structures). Moreover, by correlating the results of this study with previous reports [2,3], these two criteria are also the only ones adequate for the study of PDL, dental pulp, and NVB (with Tresca being more adequate due to criteria designed for ductile non-homogenous materials with a brittle flow mode).

Thus, based on the above, a single criterion (the Tresca) can be identified as more adequate for the FEA analysis of dental structures (i.e., PDL, dentin, enamel, cementum, dental pulp, and NVB). This type of correlation between studies (herein and previous [2,3]) also provides quantitative data about the absorption and dissipation of stress in the tooth and the surrounding periodontium (Table 3 and Table 4). Based on this functional correlation, the absorption and dissipation capacity of the tooth as a unitary structure (due to dentin and cement) can easily be highlighted (86.66–97.5% of stresses dissipated before reaching the PDL, 97.47–98.1% of stresses before reaching the NVB, and 99.6–99.94% of stresses before reaching the pulp, Tresca criteria) in Table 3. For the Von Mises criteria (Table 4), the absorption–dissipation capacity remains similar: 85.31–97.5% of stresses dissipated before reaching the PDL, 97.56–98.25% of stresses before reaching the NVB, and 99.59–99.94% of stresses before reaching the pulp. The high absorption and dissipation capacity of tooth structures (i.e., dentin and cement) was found also in the case of the other three failure criteria (S1, S3, pressure), by performing the same association with the quantitative stresses provided for PDL and dental pulp NVB by previous studies [2,3], confirming the absorption–dissipation capacity and the ductility of the tooth structure. Thus, for Tresca criteria, in the intact periodontium (0.5 N/approx. 50 g of force), out of the total amount of quantitative stress displayed by the tooth structure only a small percentage reaches the PDL (2.51–4.08% the apical third and 5.45–12.94% the cervical third (2.51–4.05% and 5.42–13.27% for Von Mises)), and an even smaller percentage the dental pulp with its NVB (1.9–2.17% NVB, and 0.07–0.4% coronal pulp (1.75–2.39% and 0.1–0.41% for VM)). However, in an 8 mm reduced periodontium, a slight increase in percentage was visible for both the PDL (3.69–6.03% for the apical third and 9.32–14.07% for the cervical third (3.85–5.59% and 9.28–14.69% for Von Mises)), and the dental pulp and NVB (2.06–2.53% for NVB and 0.06–0.21% for the coronal pulp (1.77–2.44% and 0.06–0.21% for VM)). Based of herein results, it seems that from a total of approximately 0.5 N/approx. 50 g of force, only a maximum of 0.125 N/approx. 12.5 gf reaches the PDL and a maximum of 0.01 N/approx. 1 gf the NVB and dental pulp, independently of the level of bone loss.

When comparing the Tresca with the Von Mises quantitative results (Table 2), the average numerical values were found to be within the acknowledged mathematical interval of approx. 15–30% higher (i.e., approx. 14.6% higher for the present results).

## 4. Discussion

The present study examined the five most used failure criteria for the study of the tooth structure, aiming to find the most adequate in supplying both qualitative and quantitative correct results. Additionally, evaluated the stress of the absorption–dissipation capacity of the tooth structure for all five movements when subjected to 0.5 N/approx. 50 gf of orthodontic load during 0–8 mm of periodontal breakdown.

Each failure criterion used in the FEA analysis is mathematically designed to provide correct results only in case of a certain type of material (due to various different types of yielding behaviors of materials). Thus, the S1 and S3 criteria are specific for brittle materials, where there is almost no deformation before cracking and fracture, while in pressure criteria, there are no shear stresses and it is specific for gas and liquids [2,3]. For ductile materials, where there are various plastic/elastic deformations before fracture or destruction, Tresca (for non-homogenous) and Von Mises (for homogenous) are specific [2,3]. In certain situations, like those in living tissues, Tresca is more constraining and particularly more suited than Von Mises. It must be pointed out that no FEA studies using the Tresca criteria were found for the study of the tooth; thus, the only possible correlations between herein and other studies were performed with Von Mises criteria (similar with Tresca qualitatively, but 15–30% lower quantitatively).

The tooth structure consists of three distinct materials (enamel, dentin, and cement) in different quantities (%). The dentin and cement, due to the anatomical micro-structure and physical properties, are considered ductile materials with a certain brittle flow mode that ensures the absorption and dissipation of the stress determined by functional/parafunctional/orthodontic loads [1,2,5,8]. The enamel is a brittle material that cracks and fractures under loads, transmitting the loads directly to the other two components [1,5,6]. However, out of the total volume of the dental structure, enamel represents an extremely small percentage, with the vast majority being represented by dentin and a small quantity of cement [1,5,6]. Thus, the entire structure of the tooth naturally resembles a non-homogenous ductile material with a certain brittle flow mode (which absorbs and dissipates the stresses) [1,2,5,8], confirmed by the herein stress distribution display of the Tresca and VM criteria (Table 2, Table 3 and Table 4, Figure 2, Figure 3, Figure 4, Figure 5 and Figure 6).

Nonetheless, all the previous FEA studies [10,11,12,13,14,15,17,18,19,20,21,22,23,24,25,26,27,28] of the tooth neither acknowledged nor addressed these issues when employing certain failure criteria or discussing their results. Herein study, to the best of our knowledge, is the only one of this type (i.e., comparing all five failure criteria in multiple 3D models under similar boundary conditions and addressing the material-type issues). As a result, both the Von Mises and Tresca failure criteria had been found to provide both qualitative and quantitative correct results, with Tresca being more adequate due to its algorithm-based design for non-homogenous structures. Herein report is in agreement with two recent reports (0.5 N/50 gf, five orthodontic movements, 0–8 mm periodontal breakdown, 81 models of the second lower premolar, 405 FEA simulations, five failure criteria, and similar boundary conditions) addressing the same issue of material type vs. failure criteria but with a focus on PDL and dental pulp and its NVB [2,3]. These studies [2,3] reported both criteria (Von Mises and Tresca) to be adequate for the study of PDL, dental pulp, and NVB, but with Tresca giving more accurate information. Thus, by correlating herein and the previous two reports [2,3], a single FEA criterion seems to be more suited for the study of the tooth and the surrounding PDL, providing data in agreement with in vivo knowledge [2,3].

Another issue addressed by the herein simulations was the study of stress absorption–dissipation acknowledged ability of the tooth as a single stand structure (as it functions under everyday clinical conditions). There are a lot of studies [10,11,12,13,14,15,17,18,19,20,21,22,23,24,25,26,27,28] regarding the stress display in different components of the tooth (using various FEA criteria); however, to the best of our knowledge, no other study quantified the amount of stress transmitted from the tooth to the dental pulp and PDL. This was possible due to a correlation with the two previous reports [2,3] of PDL and dental pulp NVB. Our study found that under a small orthodontic force of 0.5 N, in both intact and reduced periodontium, the tooth structure absorbs and dissipates more than 85% of the stresses before reaching the surrounding tissues (i.e., 85.31–97.5% of the stresses is dissipated before reaching the PDL, 97.56–98.25% of the stresses before reaching the NVB, and 99.59–99.94% of the stresses before reaching the pulp, Tresca criteria) (Table 3 and Table 4). These reports are in agreement with the data provided by previous studies regarding the biomechanical properties of the tooth components [1,2,5,7,8] and clinical knowledge (0.5 N is clinically safely applied in both the intact and reduced periodontium).

Closely related qualitative and quantitative results were reported by Field et al. [15] for the tipping movement (a single 3D mandibular model of 32,812 elements, holding an incisor, canine, and the first premolar, and a single 3D model holding one canine tooth of 23,565 elements, 0.35 N of tipping (and 0.5 N of resulting load), Von Mises, S1, S3, hydrostatic pressure). The reports were: Von Mises, 60–192.6 KPa on the cervical region and outer surface of root (vs. 44.6–132 KPa in the outer surface of the root in our second premolar model); S1 33.52–132.3 KPa on the distal side of the root (vs. 51.9–124.6 KPa on the vestibular distal side in our study); S3 14.6–34.2 KPa on the mesial side of root (vs. 79.3 KPa on the lingual side in our simulation); hydrostatic pressure 0.8–32 KPa in the apical third of root (vs. 39.38 KPa in the apical third of root in this report).

Field et al. [15] reported a higher stress magnitude for the model with multiple teeth (324.5 KPa for PDL and 192.6 KPa for dentin) vs. the model with a single tooth (235.5 KPa for PDL and 87.6 KPa for dentin), and 32 KPa of the hydrostatic stress in the apical third of the root, almost double the reported physiological maximum hydrostatic pressure (MHP) of 16 KPa, signaling the high risk of orthodontic external root resorption. The reported amount of stress for PDL (235.5–324.5 KPa) also exceeded by far the MHP, and normally should determine serious periodontal loss, which clinically never happens [4]. It must be emphasized that their report signals an external apical orthodontic root resorption risk and an advanced periodontal loss under a relatively small tipping force of 0.35 N/0.5 N in the intact periodontium, which normally never happens in daily practice [4].

However, in two recent FEA studies [2,3] (by employing five failure criteria) regarding the PDL and dental pulp with its NVB, 0.5 N were reported to be safe to be applied in both the intact and 8 mm periodontal breakdown for all five orthodontic movements, with no ischemic/necrotic or resorptive risks and the PDL amount of stress lower than MHP, which is in agreement with clinical data and confirmed by Gupta et al.’s [12] report (i.e., FEA Von Mises analysis, single tooth, intact periodontium, 0.15 N of intrusion, 0.3 N of extrusion and tipping, and 1 N of translation, and amounts of stress lower than MHP) and Hemant et al.’s [19,20] (i.e., S1 and S3, single tooth, intact periodontium, 0.2–0.3 N of intrusion, 0.88–1 N of lingual torque, and stresses lower than MHP). The differences between the results of this study and Fields et al.’s [15] must come from the applied loading conditions in models with various anatomical accuracy (23,565–32,812 elements, 1.2 mm mesh size vs. 5.06–6.05 million C3D4 tetrahedral elements and a global element size of 0.08–0.116 mm for herein).

Shaw et al. [13] in another Von Mises study reported a lower amount of stress than herein (single simplified model of a maxillary central incisor with the intact periodontium, VM, qualitative results comparable with herein, quantitative results lower: PDL stress lower than MHP [0.27–2 KPa], cervical stress extrusion/intrusion 9.06–11.1 KPa, tipping 8.5–11.2 KPa, rotation 10 KPa, apical stress extrusion/intrusion 9.95 KPa, tipping 8.73–9.17 KPa, and rotation 0.84 KPa).

Higher amount of the Von Mises stress were reported by Merdji et al. [14] (single simplified model of the first lower molar, VM, intrusion 10 N, translation/tipping 3 N, qualitative results comparable with herein, and quantitative results higher: translation 20.36 MPa, intrusion 18.36 MPa, and tipping 19.62 MPa); Maravic et al. [10] (single simplified model of the second upper premolar endodontically treated in the intact periodontium, 850 N of intrusion, qualitative results comparable with herein, and quantitative results higher: coronal 50 MPa and apical 4.2 MPa); Huang et al. [16] (96 copies of a single simplified model of the first lower premolar endodontically treated, 165 N of intrusion, qualitative results comparable with herein, and quantitative results higher: coronal–cervical–apical 20 MPa).

Perez-Gonzalez et al. [11] reported that for endodontically treated teeth (as above [10,16]), the Von Mises criteria were less accurate than the S1 and S3 criteria, without addressing the issue of the type of material of the inner micro-structure of tooth and filling/post/crown/composite materials (which mostly resemble ductile materials with a brittle flow mode).

The differences between herein and above reports [10,13,14,16] are supposedly due to the anatomical accuracy of the 3D models and the amount and conditions of the applied loads.

The S1 and S3 criteria, due to their better suitability for brittle materials and a certain brittle flow mode of the tooth structure, only sometimes provide results approaching the clinical reality, as shown herein simulations and in two recent reports of our group [2,3]. Vikram et al. [17] reported in a FEA study analyzing the S1 and S3 criteria lower amounts of stress in the apical third of the root (simplified model of maxillary central incisor, 0.15–0.24 mm PDL thickness, apical third stress: 0.237 KPa for 0.35 N of extrusion, 0.067–0.0751 KPa for 0.1 N of intrusion, 0.236–0.229 KPa 0.35 N of rotation, and 0.482–0.505 KPa for 0.35 N of tipping). Reddy et al. [21] reported (the simplified maxillary central incisor model of 47,229 elements and 68,337 nodes, 150 N of intrusion, in 0%–25%–50%–75% bone loss, and S3 stress) comparable qualitative results with herein result, while the quantitative results were higher (−1.18)–(−10.93) MPa (0% loss), and (−20 KPa)–(−140 MPa) (75% loss). Jeon et al. [22] reported comparable qualitative and quantitative ([−49.4]–[−75.5] KPa translation) results (S3 stress, single simplified 3D model of the first maxillary molar of 3097 nodes and 2521 elements, intact periodontium, 0.3 N of tipping, translation, and rotation). The differences between herein and above studies [17,21,22] are due to the different model of tooth used, less anatomical accuracy, and high loads.

When employing the hydrostatic pressure criteria in ductile resemblance structures (such as tooth, PDL, and dental pulp) mixed results are shown, as in herein simulations (Figure 2, Figure 3, Figure 4, Figure 5 and Figure 6 and Table 2) and in two recent reports of our group [2,3]. Moreover, the same tendency for mixed results (contradicting in vivo and in vitro data [2,3,4,19,20]) was showed when correlating various FEA hydrostatic pressure studies [24,25,26,27,28]. Thus, Hofmann et al. [25] (hydrostatic pressure, a single 3D model of the first two rooted maxillary premolars in the intact periodontium of 165,254 elements, 3–6 N of tipping, and 0.3 mm PDL thickness) reported apical root resorption (qualitative) and quantitative stress values of 16–47 KPa in the apical third for 6 N of tipping. In another report, Hofmann et al. [24] (hydrostatic pressure, a single 3D model of the first two rooted maxillary premolar in the intact periodontium of 215,887 elements, 0.5–1 N of intrusion, 0.05–0.3 mm PDL thickness) reported very unusual both qualitative and quantitative results for such a small force, which contradicted his previous above study and the existing clinical knowledge (i.e., the massive apical third and furcation root resorption and amount of stress ranging from 3.42–80.90 KPa to 9.95 TPa). In our simulation, for both movements (Figure 3 and Figure 5) in the intact periodontium, the maximum displayed hydrostatic stress is found in the cervical third (as biomechanically should be), while the apical third stress (i.e., in the entire root) is displayed only after 4 mm of periodontal breakdown. Moreover, the quantitative results of this study for 0.5 N of load were 29.9–60.9 KPa for intrusion and 39.3 KPa for tipping.

Other studies employing hydrostatic pressure criteria for assessing the optimal force for PDL in the intact periodontium also reported contradictory results. Thus, Wu et al. [26,27,28] reported various optimal forces for the intact PDL (ranging 0.28–3.31 N) for canine, premolar, and lateral incisive, but with significant differences between the same report for the same tooth (e.g., canine: rotation 1.7–2.1 N [28] and 3.31 N [26]; extrusion 0.38–0.4 N [28] and 2.3–2.6 N [27]; premolar: rotation 2.8–2.9 N [26]), much higher than 0.6–1.2 N reported by Proffit et al. [4] (0.1–1 N), Moga et al. [2,3] (0.5 N), Hemanth et al. [19,20] (0.3–1 N).

Despite being an accurate method of study, FEA has also limitations. It must be emphasized that an FEA simulation cannot accurately reproduce the clinical conditions; thus, it is important to validate the results by establishing a correlation with in vivo and in vitro data (as above). FEA correctness depends on the employed failure criteria, the anatomical accuracy of the model and boundary conditions/physical properties.

The correct employment of the FEA criteria depends on addressing the issue of the type of material the structure is made of (ductile and/or brittle) related to its physical properties and internal anatomical micro-structure (knowing that dentine, cement, PDL, dental pulp, and NVB are of ductile resemblance [1,2,3,5,8]). Each failure criterion, apart from the suitability for a tissue type, investigates different types of stress and strain that should be taken into consideration when establishing the aim of the study. When analyzing a structure made of multiple types of tissues, the type of biomechanical behavior the structure displays must be assessed, as well as the adequacy of analyzing the structure by individual components or as a single structure, and the assessment of the uniaxial stress or the triaxial stress. The selection of the proper failure theories for ductile (Von Mises and Tresca) vs. brittle (maximum principal [Rankine theory] S1 and minimum principal S3) is mandatory because these materials fail in fundamentally different ways, while the failure theories, which usually apply to ductility, are not applicable to brittleness and vice versa (as also reported by Perez-Gonzalez et al. [11]).

Stress and strain are two fundamental concepts used to describe how a body responds to external loads. In a uniaxial load, the internal forces will develop in the interior of the material to resist the applied forces, and thus, the equilibrium will be guarded. Stress is a quantity describing the distribution of internal forces within a body as it responds to externally applied loads and it is a measure of the internal force per unit area (N/mm^2^ or Pascal). By quantifying the stress, the failure of the object/material can be predicted. Normal stress can be either tensile or compressive. Strain is the quantity that describes the deformation that occurs within a body. Normal strain can also be tensile or compressive. Hooke’s law usually applies to small strains where the relation between stress and strain is linear. Normal stress acts perpendicular to a surface.

For the shear stress, the internal forces are oriented parallel to the material’s section. Hooke’s law also applies for shear stress and shear strain. The stress state at a single point within a body will have components in both the normal and the shear directions.

Biomechanically, the failure of brittle materials is different from the failure of ductile materials. For brittle materials (e.g., dental enamel, glass, or ceramic [1,5,7]), failure occurs by fracture rather than yielding. Unlike ductile materials, brittle materials tend to have compressive strengths much larger than tensile strengths (as described in the brittle materials failure theory). To be able to assess the failure of brittle materials, the two separate ultimate strengths (i.e., for tension and compression) must be known. The modified Mohr theory (slight variation on the theory), better fits the experimental data, being one of the preferred general failure theories for brittle materials. Failure theories for brittle materials need to account for the effect of hydrostatic stresses. The maximum principal/minimum principal stress theories include large areas where their use is potentially unsafe.

The prediction of the failure of a material by comparing the stress state of the object with its material properties (e.g., the yield or ultimate strengths of the material) is obtained by performing a uniaxial test. The stress state at a point can be described using the three principal stresses (i.e., most failure theories are defined as functions of the principal stresses and the material strength). Basically, the failure occurs when the maximum or minimum principal stresses reach the yield or ultimate strengths of the solid material (the brittle theory).

Hydrostatic stresses do not cause yielding in ductile materials (e.g., steel, sponge, plastic, aluminum, brass, or rubber) but cause a change in volume, being the type of stress acting on an object submerged in liquid. For a hydrostatic stress configuration, the three principal stresses are always equal, and there are no shear stresses. For a tri-axial stress state, the hydrostatic component is the average of the three principal stresses.

The mechanism that causes the yielding of ductile materials (e.g., dentine, cement, dental pulp and NVB, PDL, bone, and steel [1,2,5,6,8]) is the shear deformation. Since there are no shear stresses for the state of hydrostatic stress, this component can be very large and still not contribute to yielding. Yielding is only caused by the stresses that cause the shape distortion (i.e., the deviatoric stresses). The deviatoric component is obtained by subtracting the hydrostatic component from each of the principal stresses. Since the failure of ductile materials only depends on the deviatoric component, the failure theory for ductile materials produces the same results regardless of where Mohr’s circle is located on the horizontal axis. The maximum principal stress S1 tensile (Rankine) and the minimum principal stress S3 compressive theory is not a good failure theory for ductile materials (due to inconsistency with the statement that yielding is independent of the hydrostatic stress). However, the only two failure theories that are consistent with this statement are the Tresca and Von Mises failure criteria (i.e., the most used failure theories for ductile materials).

In the Tresca failure criteria (maximum shear stress theory), the yielding occurs when the maximum shear stress is equal to the shear stress at yielding in a uniaxial tensile test, being consistent with the statement that hydrostatic stresses do not affect yield. This theory can also be expressed as a function of the principal stresses, instead of as a function of the shear stresses (shear stress at yielding is equal to half of the yield strength of the material).

In the Von Mises failure criteria (maximum distortion energy theory), the yielding occurs when the maximum distortion energy in a material is equal to the distortion energy at yielding in a uniaxial tensile test. The distortion energy basically forms the portion of the strain energy in a stressed element corresponding to the effect of the deviatoric stresses. The distortion energy per unit of volume can be calculated from the principal stresses. At yielding during a tensile test, the maximum principal stress is equal to the yield strength of the material, and the two other principal stresses are equal to zero. This theory considers the difference between principal stresses (being independent of the hydrostatic stress); thus, if larger than the yield strength of the material, yielding had been predicted to have occurred. The equivalent Von Mises stress is a common output of stress analysis, performed using the finite element method.

When comparing Tresca (for non-homogenous materials) and Von Mises (for homogenous materials), the Tresca yield surface lies entirely inside the von Mises surface, meaning that Tresca is more conservative, with the maximum difference between the two theories calculated to be 15–30% (Von Mises < Tresca). Von Mises is usually preferred to Tresca because, in the engineering field, it agrees better with experimental data, but Tresca is sometimes used because it is easier to apply and is more conservative. Tresca and Von Mises yield surfaces are not affected by hydrostatic stresses.

Thus, based on the above biomechanical suitability of each of the five failure criteria for a type of material (considering that the tooth components and bracket have ductile resemblance), the obvious non-homogeneity of the inner micro-structure and the biomechanical display of stresses shown in Figure 3, Figure 4, Figure 5, Figure 6 and Figure 7, the Tresca and Von Mises criteria have proven to be suitable for studying the tooth as a standing structure. The anatomical accuracy of the 3D model is closely related to the number of nodes and elements, as well as the global element size (e.g., 184 times more elements and 15 times lower element size for herein compared with Fields et al.’s [15] model for VM simulation). By using the CBCT data with a 0.075 mm voxel size, the image reconstruction software produces an extremely fine mesh with a minimum of 0.075–0.08 mm element size. However, after the reconstruction of all anatomical components, 3D model assembling, and surface smoothing, the global element size reaches 0.08–0.116 mm (confirmed by the Finite elements analysis software during the mesh verification phase). A large number of elements and nodes is usually associated with a reduced global elements size and vice versa (e.g., Fields et al.’s [15], 23,565–32,812 elements, 1.2 mm mesh size vs. 5.06–6.05 million C3D4 elements, 0.08–0.116 mm element size for herein). The anatomically irregular shape of the 3D model components imposed a discretization using tetrahedral finite elements. In the ABAQUS library, both the C3D4 (first-order linear interpolation tetrahedron element) and C3D10 (second-order quadratic interpolation tetrahedron element) are available. It is acknowledged that C3D4 elements show additional rigidity, nonetheless, can be surpassed by a fine discretization of the model. However, the use of C3D10 elements, despite their better accuracy, implies an extremely high compute power. Moreover, the anatomically irregular shape makes the type of stress/displacement continuum shell (SC6R—six-node triangular in a plane continuum shell wedge) discretization impossible, while its use is recommended in bending dominant problems.

The sample size could also influence the accuracy of the results. It must be pointed out that most FEA simulations were performed on a single simplified model of the tooth (implying a sample size of one, one model of one patient), as shown above [10,13,14,15,16,17,19,20,21,22,24,25,26,27,28], while the in vitro experiments on extracted teeth were performed on multiple teeth from various patients (e.g., Maravic et al. [10], a single root premolar for the FEA simulation and fifteen premolars for the in vitro experiments). This approach is used because it is extremely difficult to performed the 3D model reconstruction, and the FEA results are validated through in vitro experiments [10]. Herein study was performed on 81 different models of the second lower premolar with the intact and reduced periodontium in a total of 405 simulations, while the sample size was nine.

It is important that the boundary conditions and physical properties of the 3D models resemble to the anatomical reality as close as possible. The tooth and surrounding tissues possess anisotropy, non-linear elasticity, and non-homogeneity, while almost all FEA studies (above included) assume isotropy, linear elasticity, and homogeneity. However, from the biomechanical point of view, under small loads of around 1 N, all tissues display linear elasticity [2,3]. The anatomical micro-structure is non-homogenous; thus, the absorption–dissipation of stress could be higher in realty than in the FEA reports. To assume non-homogeneity, it is to imply extremely complicated equations that are simply not feasible in the dental field; thus, the correlation with in vitro/in vivo data is imperative. Regarding the linear vs. non-linear issue, Hemanth et al. [19,20] in a comparative study employing the S1 and S3 (designed for brittle) criteria in PDL (a ductile material) reported 20–50% less quantitative applied force needed for non-linearity compared to linearity. Nonetheless, no study employing the Tresca or Von Mises criteria (more adequate for dental structures) found a relation to the linear vs. non-linear issue to verify Hemanth et al.’s report. This lack of data emphasizes the need for more FEA studies regarding this type of comparative analysis of multiple failure criteria.

## 5. Conclusions

Both the Von Mises and Tresca failure criteria seem to be more adequate for the FEA study of the tooth as a structure (consisting of dentin, cement, and enamel), than the S1, S3, and hydrostatic pressure criteria;The Tresca criteria seem to supply more accurate quantitative and qualitative results than Von Mises for the study of the tooth, for both the intact and reduced periodontium;FEA simulations during a 0–8 mm periodontal breakdown process showed that under a small orthodontic force, the tooth structure seems to absorb and dissipate most of the stress produced by the orthodontic load (i.e., from a total of approximately 0.5 N/approx. 50 gf of force, only a maximum of 0.125 N/approx. 12.5 gf reaches the PDL, and a maximum of 0.01N/approx. 1 gf, the NVB and dental pulp);Quantitatively, for all five failure criteria, the rotational and translational movements produced the highest amount of stress, while the intrusion and extrusion, the lowest;Only Tresca and Von Mises displayed biomechanically correct qualitative results, while the other three criteria showed various biomechanically unusual, color-coded stress displays during the entire periodontal breakdown simulation.

## 6. Practitioner Points

For a practitioner, the quantification of the tooth stress’s absorption–dissipation ability is vital, especially when various levels of periodontal breakdown are present, and the risks of ischemia, necrosis, and resorption are increased. Herein simulations showed that from a total of approximately 0.5 N/50 gf of applied force, independently of the level of bone loss, only a maximum of 0.125 N/12.5 gf seem to reach the PDL, and a maximum of 0.01 N/1 gf, the NVB and dental pulp; the rest of them are absorbed and dissipated by the tooth structure. To reduce the risks of further tissue loss, the practitioner must also know how the stress display changes in the root under various orthodontic forces and movements when the periodontal breakdown process occurs, adapting his treatment.

For a researcher, this study helps in understanding which is failure criteria/methods are better for the study of the tooth as a structure, and that they should be closely corelated with the study of surrounding tissues. It also helps both the clinician and the researcher to better understand the FEA studies and know the advantages and the limits of each method.

## Figures and Tables

**Figure 1 ijerph-20-04133-f001:**
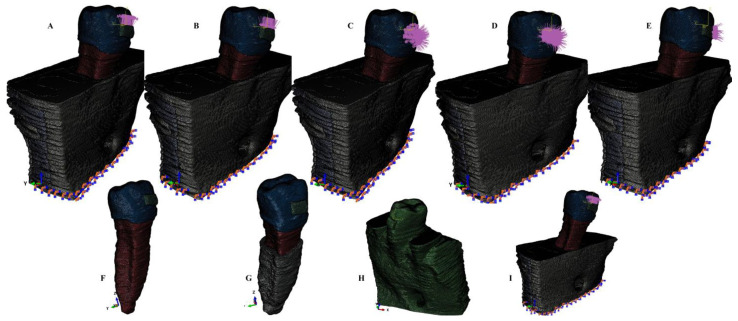
Mesh model: (**A**) 2nd lower right premolar model with 4 mm bone loss and applied vector for extrusion; (**B**) 2nd lower right premolar model with 4 mm bone loss and applied vector for intrusion; (**C**) 2nd lower right premolar model with 4 mm bone loss and applied vector for rotation; (**D**) 2nd lower right premolar model with 4 mm bone loss and applied vector for tipping; (**E**) 2nd lower right premolar model with 4 mm bone loss and applied vector for translation, (**F**) tooth model; (**G**) tooth model with 4 mm reduced PDL; (**H**) no bone loss 3D model; (**I**) 2nd lower right premolar model with 8 mm bone loss and applied vector for extrusion.

**Figure 2 ijerph-20-04133-f002:**
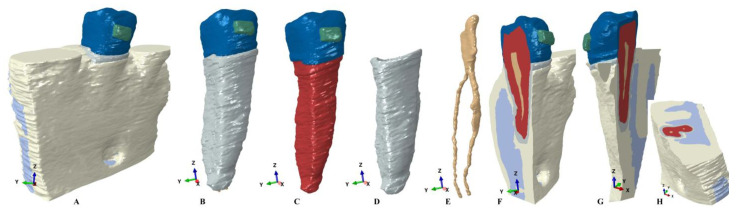
3D model without the mesh borders: (**A**) 2nd lower right premolar model with intact periodontium; (**B**) 2nd lower right premolar model with intact PDL and NVB; (**C**) 2nd lower right premolar model; (**D**) 2nd lower right premolar’s intact PDL; (**E**) 2nd lower right premolar’s dental pulp and NVB; (**F**–**H**) 3D model different sections.

**Figure 3 ijerph-20-04133-f003:**
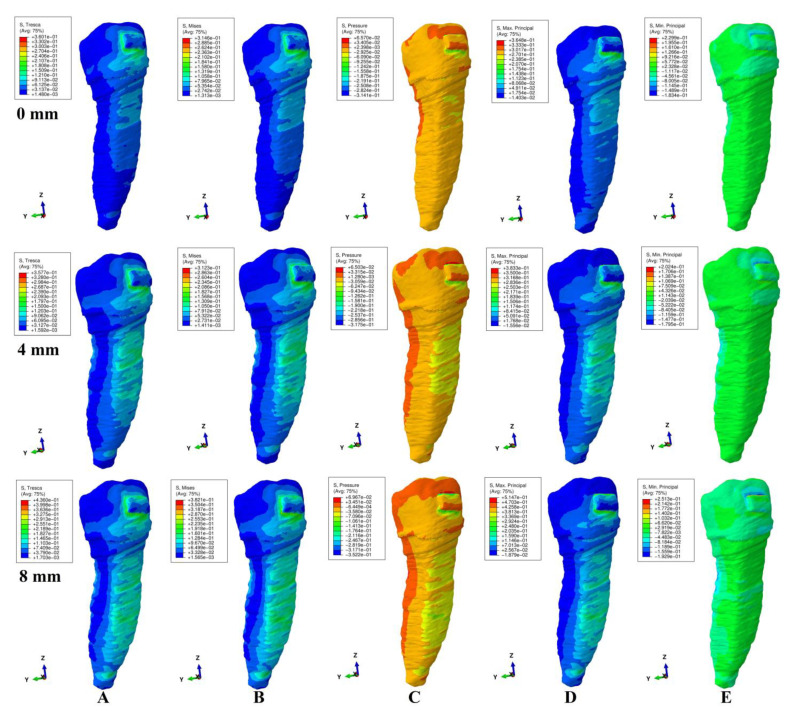
Comparative stress display of the five failure criteria in intact, 4 mm and 8 mm periodontal breakdown for the extrusion movement under 0.5 N of load: (**A**) Tresca; (**B**) Von Mises; (**C**) Pressure; (**D**) S1; (**E**) S3.

**Figure 4 ijerph-20-04133-f004:**
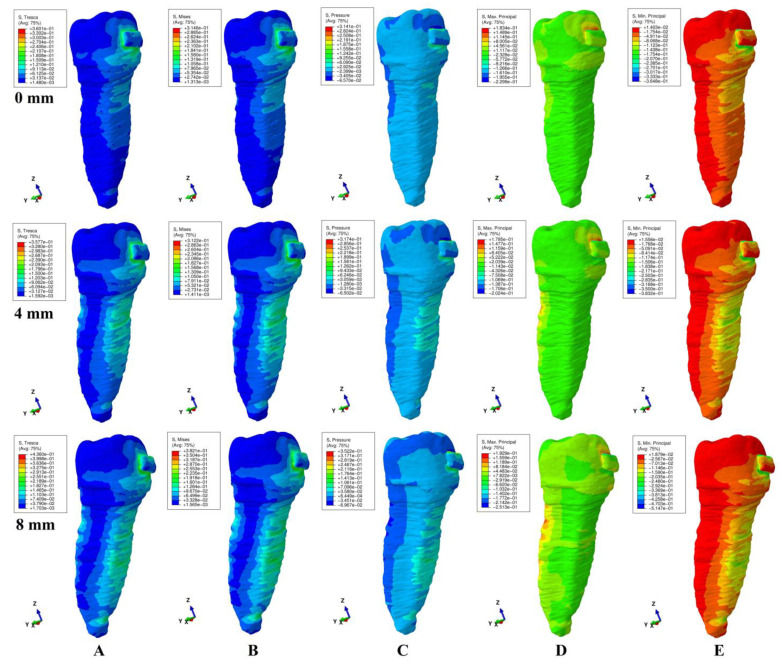
Comparative stress display of the five failure criteria in intact, 4 mm and 8 mm periodontal breakdown for the intrusion movement under 0.5 N of load: (**A**) Tresca; (**B**) Von Mises; (**C**) Pressure; (**D**) S1; (**E**) S3.

**Figure 5 ijerph-20-04133-f005:**
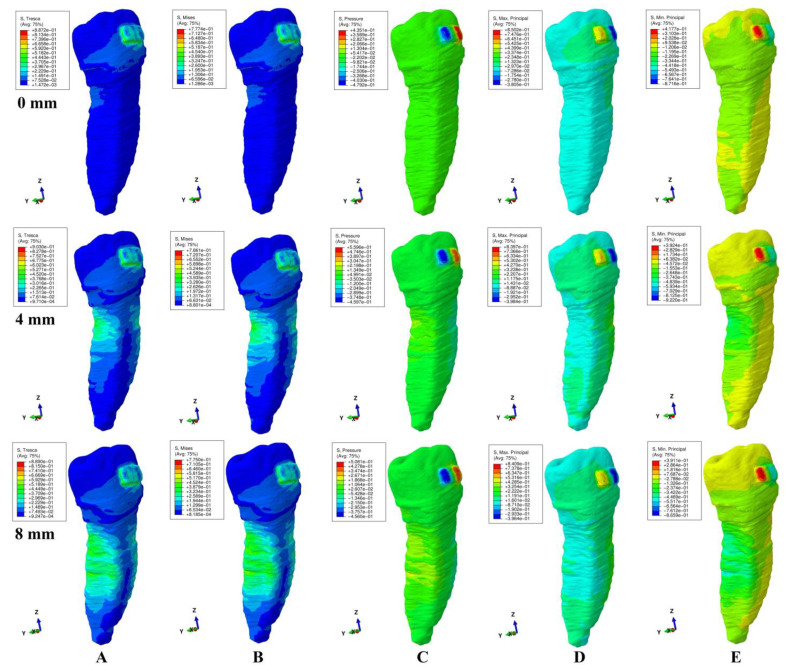
Comparative stress display of the five failure criteria in intact, 4 mm and 8 mm periodontal breakdown for the rotation movement under 0.5 N of load: (**A**) Tresca; (**B**) Von Mises; (**C**) Pressure; (**D**) S1; (**E**) S3.

**Figure 6 ijerph-20-04133-f006:**
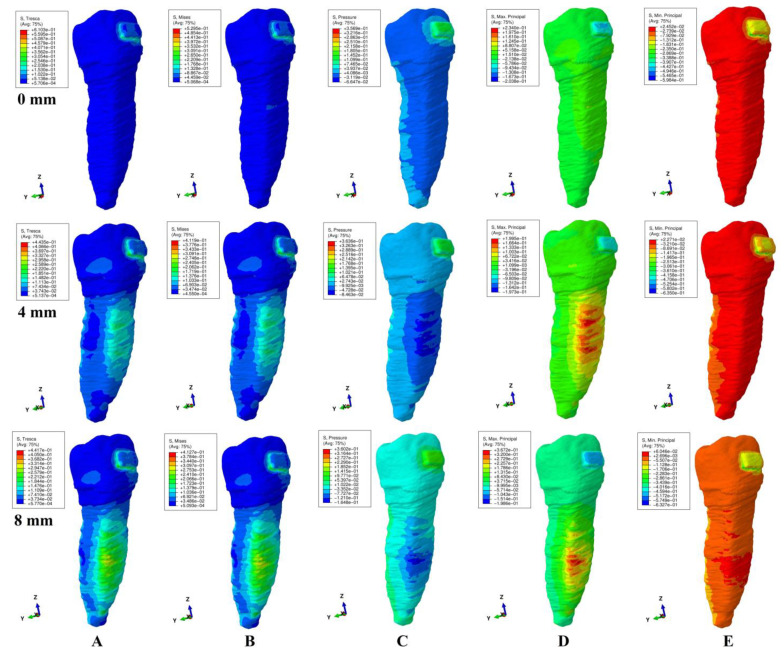
Comparative stress display of the five failure criteria in intact, 4 mm and 8 mm periodontal breakdown for the tipping movement under 0.5 N of load: (**A**) Tresca; (**B**) Von Mises; (**C**) Pressure; (**D**) S1; (**E**) S3.

**Figure 7 ijerph-20-04133-f007:**
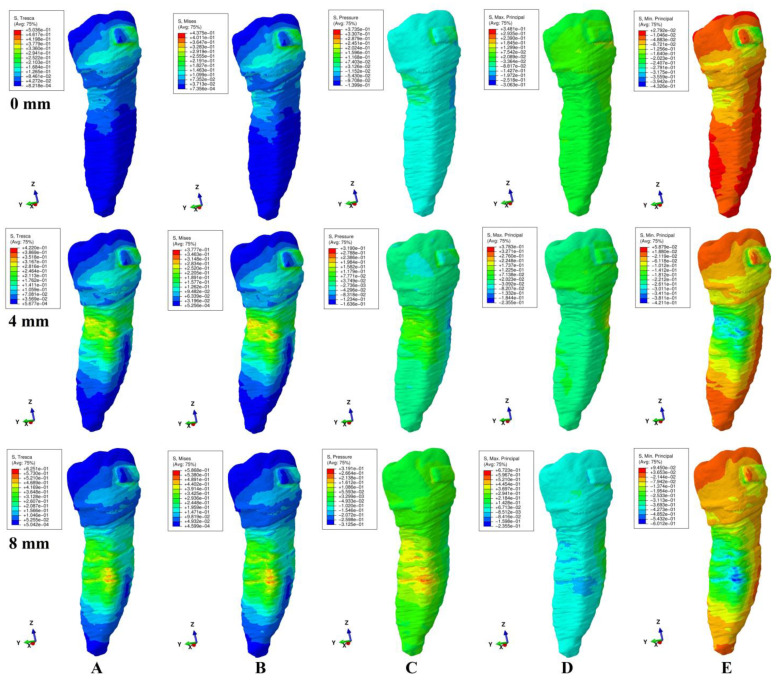
Comparative stress display of the five failure criteria in intact, 4 mm and 8 mm periodontal breakdown for the translation movement under 0.5 N of load: (**A**) Tresca; (**B**) Von Mises; (**C**) Pressure; (**D**) S1; (**E**) S3.

**Table 1 ijerph-20-04133-t001:** Elastic properties of materials table.

Material	Young’s Modulus, E (GPa)	Poisson Ratio, ʋ	Refs.
Enamel	80	0.33	[2,3]
Dentin/Cementum	18.6	0.31	[2,3]
Pulp	0.0021	0.45	[2,3]
PDL	0.0667	0.49	[2,3]
Cortical bone	14.5	0.323	[2,3]
Trabecular bone	1.37	0.3	[2,3]
Bracket (stainless steel)	190	0.265	[2,3]

**Table 2 ijerph-20-04133-t002:** Maximum stress average values (KPa) produced by 0.5 N/50 g of orthodontic forces in the tooth structure.

Resorption (mm)			0	1	2	3	4	5	6	7	8
Intrusion	**Tresca**	a	**61.25**	**68.59**	**75.94**	**83.28**	**90.62**	**104.59**	**118.56**	**132.53**	**146.50**
		m	**61.25**	**68.59**	**75.94**	**83.28**	**90.62**	**104.59**	**118.56**	**132.53**	**146.50**
		c	**91.13**	**98.42**	**105.72**	**113.01**	**120.30**	**126.85**	**133.40**	**139.95**	**146.50**
		C	**121.00**	**128.25**	**135.50**	**142.75**	**150.00**	**158.18**	**166.35**	**174.53**	**182.70**
	**VM**	a	53.54	59.92	66.33	72.78	79.11	91.43	103.76	116.08	128.40
		m	53.54	59.92	66.33	72.78	79.11	91.43	103.76	116.08	128.40
		c	79.65	85.99	92.33	98.66	105.00	110.85	116.70	122.55	128.40
		C	105.80	112.08	118.35	124.63	130.90	138.20	145.50	152.80	160.10
	Pressure	a	29.25	37.55	45.86	54.16	62.46	64.59	66.71	68.84	70.96
		m	29.25	37.55	45.86	54.16	62.46	73.37	84.28	95.19	106.10
		c	60.90	61.29	61.68	62.07	62.46	73.37	84.28	95.19	106.10
		C	60.90	69.26	77.62	85.97	94.33	97.27	100.22	103.16	106.10
	S1	a	45.61	63.18	80.76	98.33	115.90	125.90	135.90	145.90	155.90
		m	80.05	89.01	97.98	106.94	115.90	125.90	135.90	145.90	155.90
		c	114.51	130.76	147.01	163.25	179.50	182.85	186.20	189.55	192.90
		C	148.95	156.59	164.23	171.86	179.50	182.85	186.20	189.55	192.90
	S3	a	−49.11	−57.87	−66.63	−75.39	−84.15	−91.78	−99.41	−107.03	−114.66
		m	−49.11	−57.87	−66.63	−75.39	−84.15	−102.88	−121.61	−140.33	−159.06
		c	−80.68	−89.86	−99.04	−108.22	−117.40	−127.82	−138.23	−148.65	−159.06
		C	−112.30	−121.88	−131.45	−141.03	−150.60	−152.72	−154.83	−156.95	−159.06
Extrusion	**Tresca**	a	**61.25**	**68.59**	**75.94**	**83.28**	**90.62**	**104.59**	**118.56**	**132.53**	**146.50**
		m	**61.25**	**68.59**	**75.94**	**83.28**	**90.62**	**104.59**	**118.56**	**132.53**	**146.50**
		c	**91.13**	**98.42**	**105.72**	**113.01**	**120.30**	**126.85**	**133.40**	**139.95**	**146.50**
		C	**121.25**	**128.25**	**135.50**	**142.75**	**150.00**	**158.18**	**166.35**	**174.53**	**182.70**
	**VM**	a	53.54	59.92	66.33	72.78	79.11	91.43	103.76	116.08	128.40
		m	53.54	59.92	66.33	72.78	79.11	91.43	103.76	116.08	128.40
		c	79.65	85.99	92.33	98.66	105.00	110.85	116.70	122.55	128.40
		C	105.80	112.08	118.35	124.63	130.90	138.20	145.50	152.80	160.10
	Pressure	a	−29.25	−37.55	−45.86	−54.16	−62.47	−64.59	−66.71	−68.84	−70.96
		m	−29.25	−37.55	−45.86	−54.16	−62.47	−73.37	−84.28	−95.19	−106.17
		c	−60.90	−61.29	−61.68	−62.07	−62.47	−73.37	−84.28	−95.19	−106.17
		C	−60.90	−69.26	−77.62	−85.97	−94.34	−97.27	−100.22	−103.16	−106.17
	S1	a	49.11	66.19	83.27	100.34	117.42	127.82	138.21	148.61	159.00
		m	80.68	89.87	99.05	108.24	117.42	127.82	138.21	148.61	159.00
		c	112.30	113.58	114.86	116.14	117.42	127.82	138.21	148.61	159.00
		C	143.80	145.50	147.20	148.90	150.60	152.70	154.80	156.90	159.00
	S3	a	−45.61	−55.22	−64.83	−74.44	−84.05	−92.76	−101.48	−110.19	−118.90
		m	−45.61	−55.22	−64.83	−74.44	−84.05	−92.76	−101.48	−110.19	−118.90
		c	−45.61	−55.22	−64.83	−74.44	−84.05	−92.76	−101.48	−110.19	−118.90
		C	−80.05	−89.01	−97.98	−106.94	−115.90	−125.90	−135.80	145.70	−155.90
Translation	**Tresca**	a	**42.73**	**49.75**	**56.77**	**63.79**	**70.81**	**79.26**	**87.72**	**96.17**	**104.62**
		m	**84.62**	**107.52**	**130.42**	**153.31**	**176.21**	**249.40**	**322.60**	**395.79**	**468.98**
		c	**126.55**	**165.32**	**204.08**	**242.85**	**281.61**	**328.44**	**375.28**	**422.11**	**468.94**
		C	**168.40**	**187.91**	**207.42**	**226.92**	**246.43**	**250.01**	**253.59**	**257.16**	**260.74**
	**VM**	a	37.13	43.70	50.26	56.83	63.39	72.09	80.79	89.49	98.19
		m	73.52	102.43	131.34	160.25	189.16	252.38	315.60	378.81	442.03
		c	109.93	145.46	180.98	216.51	252.03	299.53	347.03	394.53	442.03
		C	146.31	164.87	183.44	202.00	220.56	226.64	232.72	238.80	244.88
	Pressure	a	31.26	32.82	34.38	35.93	37.49	68.42	99.35	130.28	161.21
		m	31.26	58.94	74.61	96.29	117.96	155.09	192.23	229.36	266.49
		c	74.03	85.01	96.00	106.98	117.96	155.09	192.23	229.36	266.49
		C	74.03	85.01	96.00	106.98	117.96	128.77	139.59	150.40	161.21
	S1	a	75.42	87.21	99.00	110.79	122.58	127.64	132.70	137.76	142.82
		m	75.42	87.21	99.00	110.79	122.58	165.48	208.38	251.28	294.18
		c	129.91	140.86	151.82	167.68	173.72	203.84	233.95	264.07	294.18
		C	129.91	140.86	151.82	167.68	173.72	203.84	233.95	264.07	294.18
	S3	a	−48.84	−51.93	−55.02	−58.10	−61.19	80.37	−99.55	−118.73	−137.91
		m	−87.21	−100.72	−114.24	−127.75	−141.26	−212.77	−284.29	−355.80	−427.31
		c	−164.09	−208.35	−252.60	−296.86	−341.11	−377.14	−413.16	−449.19	−485.21
		C	−164.09	−168.38	−172.67	−176.95	−181.24	−199.26	−217.28	−235.30	−253.32
Rotation	**Tresca**	a	**75.28**	**80.03**	**99.03**	**118.03**	**151.30**	**169.20**	**187.10**	**205.00**	**222.90**
		m	**75.28**	**131.86**	**188.44**	**245.02**	**301.60**	**355.94**	**410.29**	**464.63**	**518.97**
		c	**149.10**	**187.23**	**225.35**	**263.48**	**301.60**	**355.94**	**410.29**	**464.63**	**518.97**
		C	**222.90**	**242.58**	**262.25**	**281.93**	**301.60**	**318.94**	**336.27**	**353.61**	**370.94**
	**VM**	a	65.96	82.40	98.83	115.27	131.70	147.38	163.05	178.73	194.40
		m	65.96	115.12	164.29	213.45	262.61	310.08	357.55	405.02	452.49
		c	130.60	163.60	196.60	229.60	262.61	310.08	357.55	405.02	452.49
		C	195.31	212.14	229.02	245.90	262.61	277.82	293.04	308.25	323.46
	Pressure	a	54.17	74.37	94.58	114.78	134.98	147.94	160.90	173.85	186.81
		m	54.17	74.37	94.58	114.78	134.98	168.03	201.09	234.14	267.19
		c	130.15	135.05	139.95	144.85	149.75	179.11	208.47	237.83	267.19
		C	130.15	135.05	139.95	144.85	149.75	159.02	168.30	177.57	186.84
	S1	a	−72.86	−109.83	−146.75	−183.71	220.73	246.90	273.08	299.25	325.42
		m	−72.86	−109.83	−146.75	−183.71	220.73	246.90	273.08	299.25	325.42
		c	132.31	154.42	176.52	198.63	220.73	246.90	273.08	299.25	325.42
		C	132.31	154.42	176.52	198.63	220.73	246.90	273.08	299.25	325.42
	S3	a	−119.51	−128.47	−137.43	−146.38	−155.34	−175.87	−196.40	−216.93	−237.46
		m	−119.51	−128.47	−137.43	−146.38	−155.34	−228.25	−301.15	−374.00	−446.96
		c	−226.91	−236.40	−245.89	−255.37	−264.86	−310.36	−355.88	−401.40	−446.96
		C	−226.91	−236.40	−245.89	−255.37	−264.86	−284.21	−303.56	−322.91	−342.27
Tipping	**Tresca**	a	**51.38**	**66.38**	**81.38**	**96.37**	**111.37**	**120.45**	**129.53**	**138.60**	**147.68**
		m	**51.38**	**75.59**	**99.80**	**124.00**	**148.21**	**184.84**	**221.46**	**258.09**	**294.71**
		c	**102.21**	**113.71**	**125.21**	**136.71**	**148.21**	**184.84**	**221.46**	**258.09**	**294.71**
		C	**150.14**	**158.12**	**166.21**	**174.09**	**182.07**	**201.05**	**220.03**	**239.01**	**257.99**
	**VM**	a	44.59	59.29	73.99	88.68	103.38	112.01	120.65	129.28	137.91
		m	44.59	67.87	91.14	114.42	137.69	172.20	206.51	240.92	275.33
		c	88.67	100.93	113.18	125.44	137.69	172.20	206.51	240.92	275.33
		C	132.88	134.08	135.29	136.49	137.69	163.53	189.36	215.20	241.03
	Pressure	a	39.38	55.08	70.77	86.47	102.16	112.02	121.88	131.73	141.59
		m	39.38	55.08	70.77	86.47	102.16	112.02	121.88	131.73	141.59
		c	39.38	55.08	70.77	86.47	102.16	112.02	121.88	131.73	141.59
		C	39.38	55.08	70.77	86.47	102.16	112.02	121.88	131.73	141.59
	S1	a	51.59	63.79	76.00	88.19	100.40	119.96	139.51	159.07	178.62
		m	88.81	116.49	144.16	171.84	199.51	229.64	259.77	289.89	320.02
		c	124.59	143.32	162.05	180.78	199.51	217.86	236.21	254.55	272.90
		C	−130.81	−130.91	−131.01	−131.10	−131.20	−136.27	−141.34	−146.41	−151.48
	S3	a	−79.29	−81.87	−84.46	−87.04	−89.62	95.43	−101.25	−107.06	−112.87
		m	−79.29	−94.90	−110.50	−126.11	−141.71	−148.94	−156.16	−163.39	−170.61
		c	−79.29	−94.90	−110.50	−126.11	−141.71	−148.94	−156.16	−163.39	−170.61
		C	−131.21	−133.84	−136.46	−139.09	−141.71	−148.94	−156.16	−163.39	−170.61

a—apical third; m—middle third; c—cervical third; C—coronal/crown.

**Table 3 ijerph-20-04133-t003:** Tresca criteria—maximum stress average values (KPa) produced by 0.5 N/50 g of force in tooth, PDL, NVB pulp, and % of the tooth quantitative stress that is displayed by the PDL, dental pulp, and NVB (absorption–dissipation).

Resorption	mm		Apical	Middle	Cervical	Coronal	% Apical	% Middle	% Cervical	% Coronal
Tooth	0	rotation	75.28	75.28	149.10	222.90	100.00	100.00	100.00	100.00
	8	rotation	222.90	518.97	518.97	370.94	100.00	100.00	100.00	100.00
	0	translation	42.73	84.62	126.55	168.40	100.00	100.00	100.00	100.00
	8	translation	104.62	468.98	468.94	260.74	100.00	100.00	100.00	100.00
	0	tipping	51.38	51.38	102.21	150.14	100.00	100.00	100.00	100.00
	8	tipping	147.68	294.71	294.71	257.99	100.00	100.00	100.00	100.00
	0	intrusion	61.25	61.25	91.13	121.00	100.00	100.00	100.00	100.00
	8	intrusion	146.50	146.50	146.50	182.70	100.00	100.00	100.00	100.00
	0	extrusion	61.25	61.25	91.13	121.25	100.00	100.00	100.00	100.00
	8	extrusion	146.50	146.50	146.50	182.70	100.00	100.00	100.00	100.00
PDL	0	rotation	1.94	3.85	17.17		2.58	5.11	11.52	
Ref. [3]	8	rotation	8.23	16.30	73.00		3.69	3.14	14.07	
	0	translation	1.68	3.31	16.37		3.92	3.91	12.94	
	8	translation	6.31	12.51	62.57		6.03	2.67	13.34	
	0	tipping	1.29	2.55	11.36		2.51	4.96	11.11	
	8	tipping	6.89	10.28	37.42		4.66	3.49	12.70	
	0	intrusion	2.50	2.50	4.97		4.08	4.08	5.45	
	8	intrusion	6.85	6.85	13.66		4.68	4.68	9.32	
	0	extrusion	2.50	2.50	5.59		4.08	4.08	6.13	
	8	extrusion	6.87	6.87	18.75		4.69	4.69	12.80	
NVB Pulp	0	rotation	1.43	0.25	0.25	0.25	1.90	0.33	0.17	0.11
Ref. [2]	8	rotation	5.11	0.47	0.47	0.47	2.29	0.09	0.09	0.13
	0	translation	0.93	0.08	0.08	0.16	2.17	0.10	0.07	0.09
	8	translation	2.15	0.30	0.30	0.30	2.06	0.06	0.06	0.12
	0	tipping	1.22	0.21	0.21	0.21	2.38	0.40	0.20	0.14
	8	tipping	3.74	0.32	0.32	0.32	2.53	0.11	0.11	0.13
	0	intrusion	1.16	0.12	0.12	0.12	1.89	0.20	0.14	0.10
	8	intrusion	3.62	0.31	0.31	0.31	2.47	0.21	0.21	0.17
	0	extrusion	1.20	0.12	0.12	0.12	1.96	0.20	0.14	0.10
	8	extrusion	3.62	0.31	0.31	0.31	2.47	0.21	0.21	0.17

Apical—apical third; middle—middle third; cervical—cervical third, coronal; % Apical—% in apical third; % middle—% in middle third; % cervical—% in cervical third; % coronal—% in crown.

**Table 4 ijerph-20-04133-t004:** VM criteria—maximum stress average values (KPa) produced by 0.5 N/50 g of force in tooth, PDL, NVB pulp, and % of the tooth quantitative stress that is displayed by the PDL, dental pulp and NVB (absorption–dissipation).

Resorption	mm		Apical	Middle	Cervical	Coronal	% Apical	% Middle	% Cervical	% Coronal
Tooth	0	rotation	65.96	65.96	130.60	195.31	100.00	100.00	100.00	100.00
	8	rotation	194.40	452.49	452.49	323.46	100.00	100.00	100.00	100.00
	0	translation	37.13	73.52	109.93	146.31	100.00	100.00	100.00	100.00
	8	translation	98.19	442.03	442.03	244.88	100.00	100.00	100.00	100.00
	0	tipping	44.59	44.59	88.67	132.88	100.00	100.00	100.00	100.00
	8	tipping	137.91	275.33	275.33	241.03	100.00	100.00	100.00	100.00
	0	intrusion	53.54	53.54	79.65	105.80	100.00	100.00	100.00	100.00
	8	intrusion	128.40	128.40	128.40	160.10	100.00	100.00	100.00	100.00
	0	extrusion	53.54	53.54	79.65	105.80	100.00	100.00	100.00	100.00
	8	extrusion	128.40	128.40	128.40	160.10	100.00	100.00	100.00	100.00
PDL	0	rotation	1.68	3.33	14.80		2.55	5.05	11.33	
Ref. [3]	8	rotation	7.49	14.86	66.46		3.85	3.28	14.69	
	0	translation	1.49	2.95	14.59		4.02	4.01	13.27	
	8	translation	5.49	10.93	54.44		5.59	2.47	12.32	
	0	tipping	1.12	2.21	9.85		2.51	4.96	11.10	
	8	tipping	6.07	9.07	33.00		4.40	3.29	11.99	
	0	intrusion	2.17	2.17	4.32		4.05	4.05	5.42	
	8	intrusion	6.00	6.00	11.91		4.67	4.67	9.28	
	0	extrusion	2.17	2.17	4.85		4.05	4.05	6.09	
	8	extrusion	6.00	6.00	16.34		4.67	4.67	12.73	
NVB Pulp	0	rotation	1.25	0.11	0.22	0.22	1.90	0.17	0.17	0.16
Ref. [2]	8	rotation	3.44	0.45	0.45	0.45	1.77	0.10	0.10	0.10
	0	translation	0.81	0.07	0.21	0.21	2.17	0.10	0.10	0.19
	8	translation	1.95	0.19	0.26	0.26	1.99	0.04	0.04	0.06
	0	tipping	1.06	0.18	0.18	0.18	2.39	0.41	0.41	0.20
	8	tipping	3.30	0.29	0.29	0.29	2.39	0.10	0.10	0.10
	0	intrusion	0.94	0.11	0.11	0.11	1.75	0.21	0.21	0.14
	8	intrusion	3.14	0.27	0.27	0.27	2.44	0.21	0.21	0.21
	0	extrusion	0.94	0.11	0.11	0.11	1.75	0.21	0.21	0.14
	8	extrusion	3.14	0.27	0.27	0.27	2.44	0.21	0.21	0.21

Apical—apical third; middle—middle third; cervical—cervical third, coronal; % Apical—% in apical third; % middle—% in middle third; % cervical—% in cervical third; % coronal—% in crown.

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
