# Peer review of "Finite Elements Analysis of Tooth—A Comparative Analysis of Multiple Failure Criteria"

_ijerph, 2023, doi:10.3390/ijerph20054133_

Round 1

Reviewer 1 Report

The article is very interesting and I would like to congratulate the authors.

However, some issues should be addressed before publishing:

In several parts of the paper, the loads are described in grams of force? The International System of Units uses Newtons, and grams force is gf, not g. This should be addressed, I recomend use N, and if the authors want the grams force put both of them for clarity.

In introduction, the brackets are usually made of Stainless Steel, not steel. This should be corrected.

Additionally, the simulation uses Co-Cr alloys, why not Stainless steel. Is these alloys also used? If so, complete the introduction also with these alloys.

Table 1. I suppose that the alloy employed is made of Co-Cr alloy, not Cr-Co alloys. The First ones are cobalt based alloys, typically used in dentistry.

Figure 1. It is possible to have the figures without the black borders of the mesh? It would be more easy to see the models.

Can you add a figure where at least one part section have all different materials identified. It would be good for understanding the inner structure of the tooth evaluated.

Author Response

Department of Cariology, Endodontics and Oral Pathology

Faculty of Dental Medicine

University of Medicine and Pharmacy

Ms. Queenie Li

Section Managing Editor

Special Issue- Advances of Digital Dentistry and Prosthodontics 

                                                                                                                            February 19, 2023

Dear Ms. Queenie Li,

Thank you very much for your letter dated February 14, 2023, with the comments of the reviewers. We have now carefully considered the comments of the reviewers and amended the paper accordingly. All changes are highlighted in red throughout the manuscript and included also below.

Reply to Reviewer #1:

We agree and we thank the reviewer for his/her time and comments. Appropriate changes in the manuscript have by now been made. Please see below and in the manuscript.

Concern of the reviewer:

”The article is very interesting, and I would like to congratulate the authors.

However, some issues should be addressed before publishing:

In several parts of the paper, the loads are described in grams of force. The International System of Units uses Newtons, and grams force is gf, not g. This should be addressed, I recommend use N, and if the authors want the grams force put both of them for clarity.

In introduction, the brackets are usually made of Stainless Steel, not steel. This should be corrected.

Additionally, the simulation uses Co-Cr alloys, why not Stainless steel. Are these alloys also used? If so, complete the introduction also with these alloys.

Table 1. I suppose that the alloy employed is made of Co-Cr alloy, not Cr-Co alloys. The First ones are cobalt based alloys, typically used in dentistry.

Figure 1. It is possible to have the figures without the black borders of the mesh ? It would be more easy to see the models.

Can you add a figure where at least one part section have all different materials identified. It would be good for understanding the inner structure of the tooth evaluated.”

Point-by-point response to the reviewer’s comments:

  1. Concern of the reviewer:

“In several parts of the paper, the loads are described in grams of force. The International System of Units uses Newtons, and grams force is gf, not g. This should be addressed, I recommend use N, and if the authors want the grams force put both of them for clarity.”

Our response:

  • We thank the reviewer for his/her concern and comments. We do hope that our changes are according to the reviewer‘s remarks.

Revised text: The problem of the correction of the loads was performed. We followed reviewer suggestion and added both Newtons, and grams force. Please see the entire manuscript.

  1. Concern of the reviewer:

“In introduction, the brackets are usually made of Stainless Steel, not steel. This should be corrected.

Additionally, the simulation uses Co-Cr alloys, why not Stainless steel. Are these alloys also used? If so, complete the introduction also with these alloys.

Table 1. I suppose that the alloy employed is made of Co-Cr alloy, not Cr-Co alloys. The First ones are cobalt based alloys, typically used in dentistry.”

Our response:

  • We thank the reviewer for his/her concern and comments. We do hope that our changes are according to the reviewer‘s remarks.

The material type of the bracket was the stainless steel. The simulations were performed using the stainless-steel values showed in Table I (which was also corrected).

Revised text: pg.1 line 43 and Table I

“The bracket is usually made from stainless steel (a ductile material) and cemented to the enamel.”

The problem of the correction of the loads was performed. We followed reviewer suggestion and added both Newtons, and grams force. Please see the entire manuscript.

  1. Concern of the reviewer:

“Figure 1. It is possible to have the figures without the black borders of the mesh ? It would be more easy to see the models.

Can you add a figure where at least one part section have all different materials identified. It would be good for understanding the inner structure of the tooth evaluated.”

Our response:

  • We thank the reviewer for his/her concern and comments. We do hope that our changes are according to the reviewer‘s remarks.

The Figure 1 was intended to expressly represent the models with mesh borders in order to show the anatomical accuracy (high number of elements and nodes). For a better view of the models the Figure 2 was added as reviewer suggested.

Revised text: Please see Figure 2

Reviewer 2 Report

1. L49. It should be From the biomechanical point of view, not Form

2. L56. Under the enamel component. I think inner may be more accurate.

3. L102. Each method of study of the tooth..

How about Each method of studying tooth?

4. L105. simulated neither the clinical situation and conditions, not reproduce

5. Figure 1. Those pictures are too dark. Can you use the vertical profile of each model to display the mesh.

6. Before analyzing the model data obtained from CBCT tomography, it is recommended to smooth its surface first, which will not only help stress transmission more uniform, but also avoid the singularity. Then those pictures of models can be more artistic.

7. Please show the evidence of determining the mesh size of the models and how to judge the mesh quality is appropriate in this experiment.

8. Please focus on comparing the advantages and disadvantages of the five failure criteria in the discussion, and give a more detailed and clear explanation about how to get the conclusion.

Author Response

Department of Cariology, Endodontics and Oral Pathology

Faculty of Dental Medicine

University of Medicine and Pharmacy

Ms. Queenie Li

Section Managing Editor

Special Issue- Advances of Digital Dentistry and Prosthodontics 

                                                                                                                            February 19, 2023

Dear Ms. Queenie Li,

Thank you very much for your letter dated February 14, 2023, with the comments of the reviewers. We have now carefully considered the comments of the reviewers and amended the paper accordingly. All changes are highlighted in red throughout the manuscript and included also below.

Reply to Reviewer #2:

We agree and we thank the reviewer for his/her time and comments. Appropriate changes in the manuscript have by now been made. Please see below and in the manuscript.

Concerns of the reviewer:

” 1. L49. It should be “From the biomechanical point of view”, not Form

  1. L56. “Under the enamel component”. I think inner may be more accurate.
  2. L102. Each method of study of the tooth..

How about Each method of studying tooth?

  1. L105. simulated neither the clinical situation and conditions, not reproduce
  2. Figure 1. Those pictures are too dark. Can you use the vertical profile of each model to display the mesh.
  3. Before analyzing the model data obtained from CBCT tomography, it is recommended to smooth its surface first, which will not only help stress transmission more uniform, but also avoid the singularity. Then those pictures of models can be more artistic.
  4. Please show the evidence of determining the mesh size of the models and how to judge the mesh quality is appropriate in this experiment.
  5. Please focus on comparing the advantages and disadvantages of the five failure criteria in the discussion, and give a more detailed and clear explanation about how to get the conclusion.”

Point-by-point response to the reviewer’s comments:

  1. Concern of the reviewer:

“1. L49. It should be “From the biomechanical point of view”, not Form

  1. L56. “Under the enamel component”. I think inner may be more accurate.
  2. L102. Each method of study of the tooth..

How about Each method of studying tooth?

  1. L105. simulated neither the clinical situation and conditions, not reproduce.”

Our response:

  • We thank the reviewer for his/her concern and comments. We do hope that our changes are according to the reviewer‘s remarks.

Revised text:  pg.2 lines: 49, 56/ pg.3 lines: 103, 105-107

“From the biomechanical point of view enamel…”

“Inner the enamel component…”

“Each method of studying the tooth and its …”

“Only FEA studies can individually assess each component’ behavior but cannot accurately simulate neither the clinical situation and conditions, nor the micro-architecture of tissues…”

  1. Concern of the reviewer:

“5. Figure 1. Those pictures are too dark. Can you use the vertical profile of each model to display the mesh.”

Our response:

  • We thank the reviewer for his/her concern and comments. We do hope that our changes are according to the reviewer‘s remarks.

The Figure 1 was intended to expressly represent the models with mesh borders in order to show the anatomical accuracy (high number of elements and nodes). However, for a better view of the models the Figure 2 was added as reviewer suggested.

  1. Concern of the reviewer:

“6. Before analyzing the model data obtained from CBCT tomography, it is recommended to smooth its surface first, which will not only help stress transmission more uniform, but also avoid the singularity. Then those pictures of models can be more artistic.

  1. Please show the evidence of determining the mesh size of the models and how to judge the mesh quality is appropriate in this experiment.”

Our response:

  • We thank the reviewer for his/her concern and comments. We do hope that our changes are according to the reviewer‘s remarks.

Revised text: pg. 4-5, lines: 194-208

“As a downside of the manual reconstruction technique, a limited number of surface anomalies/irregularities had been present in all models, but with quasi-continuity in all areas affected by stress, the internal processes (i.e., algorithm based) have been passed. The mesh convergence testing was performed for all models. Both software (AMIRA and ABAQUS) prevents the mesh creation and the FEA analysis (i.e., due to internal algorithms) if many surface anomalies are present, allowing only a limited number of anomalies that do not interfere with the process. Nonetheless, a limited number of surface anomalies and irregularities was expected to remain even after several mesh smoothing processes but without interfering with the analysis or results (e.g., the models presented in Figure 1 and 2 displayed a total number of 264 element warnings [39 element warnings for the 665501 elements of the tooth with bracket] from a total of 6.05 million elements, and no errors). All the element warnings were in areas where the stress concentrations are reduced while areas with stress concentrations are quasi-continuous, thus the accuracy of results was not altered.”

                     Pg.22-23, lines: 648-667

“The anatomical accuracy of the 3D model is closely related to the number of nodes and elements as well as the global element size (e.g., 184 times more elements and 15 times lower element size for herein, compared with Fields et al. [15] model for VM simulation). By using a CBCT data with a 0.075 mm voxel size, the image reconstruction software produces an extremely fine mesh with a minimum of 0.075-0.08 mm element size. However, after the reconstruction of all anatomical components, 3D models assembling and the surface smoothing, the global element size reaches 0.08-0.116 mm (confirmed by the finite elements analysis software during the mesh verify phase). A large number of elements and nodes usually associates with a reduced global elements size and vice-versa (e.g., Fields’s et al. [15], 23565-32812 elements, 1.2 mm mesh size vs. 5.06-6.05 million C3D4 elements, 0.08-0.116 mm element size for herein). Anatomical irregular shape of the 3D model components imposed a discretization using of tetrahedral finite elements. In ABAQUS library are available both C3D4 (first order linear interpolation tetrahedron element) and C3D10 (second order quadratic interpolation tetrahedron element). It is acknowledged that C3D4 elements show additional rigidity, nonetheless, can be sur-passed by a fine discretization of the model. However, the use of C3D10 elements despite their better accuracy imply an extremely high compute power. Moreover, the anatomical irregular shape makes impossible the type of stress/displacement continuum shell (SC6R - six node triangular in plane continuum shell wedge) discretization while their use is recommended in bending dominated problems.”

  1. Concern of the reviewer:

“8. Please focus on comparing the advantages and disadvantages of the five failure criteria in the discussion, and give a more detailed and clear explanation about how to get the conclusion.”

Our response:

  • We thank the reviewer for his/her concern and comments. We do hope that our changes are according to the reviewer‘s remarks.

Revised text: pg. 21-26, lines: 556-667

“The correct employment of FEA criteria depends on addressing the issue of the type of material the structure is made of (ductile and/or brittle) related to its physical proper-ties and internal anatomical micro-structure (knowing that dentine, cement, PDL, dental pulp and NVB being of ductile resemblance [1-3, 5, 8]). Each failure criteria apart the suitability for a tissue type, investigates different types of stress and strain that should be taken into consideration when establishing the aim of the study. When analyzing a structure made of multiple types of tissues, it must be assessed the type of biomechanical behavior the structure displays, the adequacy of analyzing the structure by individual components or as a single structure, and the assessment of the uniaxial stress or the tri-axial stress. The selection of the proper failure theories for ductile (Von Mises and Tresca) vs. brittle (Maximum Principal [Rankine theory] S1 and Minimum Principal S3) is mandatory because these materials fail in fundamentally different, while the failure theories which apply for ductility usually are not applicable for brittle and vice-versa (as also re-ported by Perez-Gonzalez et al. [11]).

Stress and strain are two fundamental concepts used to describe how a body responds to external loads. In a uniaxial load the internal forces will develop in the interior of the material to resist to the applied forces and thus the equilibrium will be guarded. Stress is a quantity describing the distribution of internal forces within a body as it responds to externally applied loads and being a measure of the internal force per unit area (N/mm2 or Pascal). By quantifying the stress, the fail of the object/material can be predicted. Normal stress can be either tensile or compressive. Strain is the quantity that de-scribes the deformation that occur within a body. Normal strain can also be tensile or compressive. Hooke’s law usually applies for small strains where the relation between stress and strain is linear. Normal stress is acting perpendicular to a surface.

For the shear stress the internal forces are oriented parallel to the material's section. Hooke’s law also applies for shear stress and shear strain. The stress state at a single point within a body will have components in both the normal and the shear directions.

Biomechanically the failure of brittle materials is different from the failure of ductile materials. For brittle materials (e.g., dental enamel, glass, or ceramic [1, 5, 7]), failure occurs by fracture rather than yielding. Unlike ductile materials, brittle materials tend to have compressive strengths much larger than tensile strengths (being described in the brittle materials failure theory). For being able to assess the failure of brittle materials the two separate ultimate strengths (i.e., for tension and compression) must be known. The modified Mohr theory (slight variation on the theory) better fits experimental data, being one of the preferred general failure theories for brittle materials. Failure theories for brittle materials need to account for the effect of hydrostatic stresses. The maximum principal/minimum principal stress theory have large areas where their use is potentially unsafe.

To predict the failure of a material by comparing the stress state in the object with material properties (e.g., the yield or ultimate strengths of the material), are obtained by performing a uniaxial test. The stress state at a point can be described using the three principal stresses (i.e., most failure theories are defined as a function of the principal stresses and the material strength). Basically, the failure occurs when the maximum or minimum principal stresses reach the yield or ultimate strengths of the solid material (the brittle theory).

Hydrostatic stresses do not cause yielding in ductile materials (e.g., steel, sponge, plastic, aluminum, brass, or rubber) but causes a change in volume, being the type of stress acting on an object submerged in liquid. For a hydrostatic stress configuration, the three principal stresses are always equal, and there are no shear stresses. For a tri-axial stress state the hydrostatic component is the average of the three principal stresses.

The mechanism that causes yielding of ductile materials (e.g., dentine, cement, dental pulp and NVB, PDL, bone, steel [1, 2, 5, 6, 8]) is the shear deformation. Since there are no shear stresses for a state of hydrostatic stress, this component can be very large and still not contribute to yielding. Yielding is only caused by the stresses which cause the shape distortion (i.e., the deviatoric stresses). The deviatoric component is obtained by subtracting the hydrostatic component from each of the principal stresses. Since failure of ductile materials only depends on the deviatoric component, the failure theory for ductile materials produces the same results regardless of where Mohr’s circle is located on the horizontal axis. The maximum principal stress S1 tensile (Rankine) and minimum principal stress S3 compressive theory is not a good failure theory for ductile materials (due to inconsistency with the statement that yielding is independent of the hydrostatic stress). However, the only two failure theories which are consistent with this statement are Tresca and Von Mises failure criteria (i.e., the most used failure theories for ductile materials).

In Tresca failure criteria (maximum shear stress theory) the yielding occurs when the maximum shear stress is equal to the shear stress at yielding in a uniaxial tensile test, being consistent with the statement that hydrostatic stresses do not affects yield. This theory can also be expressed as a function of the principal stresses, instead of as a function of the shear stresses (shear stress at yielding is equal to half of the yield strength of the material).

In Von Mises failure criteria (maximum distortion energy theory) the yielding occurs when the maximum distortion energy in a material is equal to the distortion energy at yielding in a uniaxial tensile test. The distortion energy basically forms the portion of strain energy in a stressed element corresponding to the effect of the deviatoric stresses. The distortion energy per unit of volume can be calculated from the principal stresses. At yielding during a tensile test the maximum principal stress is equal to the yield strength of the material, and the two other principal stresses are equal to zero. This theory considers the difference between principal stresses, (being independent of the hydrostatic stress), thus, if larger than the yield strength of the material, yielding is predicted to have occurred. The equivalent Von Mises stress is a common output from stress analysis per-formed using the finite element method.

When comparing Tresca (for non-homogenous materials) and Von Mises (for homogenous materials), Tresca yield surface lies entirely inside the von Mises surface, meaning that Tresca is more conservative, with the maximum difference between the two theories calculated to be 15-30% (Von Mises < Tresca). Von Mises is usually preferred to Tresca because in the engineering field agrees better with experimental data, but Tresca is sometimes used because is easier to apply and more conservative. Tresca and Von Mises yield surface are not affected by hydrostatic stresses.

Thus, based on the above biomechanical suitability of each of the five failure criteria for a type of material (considering that the tooth components and bracket have ductile resemblance), the obvious non-homogeneity of inner micro-structure and the biomechanical display of stresses showed in Figures 3-7, Tresca and Von Mises criterion have proven to be suitable for studying the tooth a standing structure.

The anatomical accuracy of the 3D model is closely related to the number of nodes and elements as well as the global element size (e.g., 184 times more elements and 15 times lower element size for herein, compared with Fields et al. [15] model for VM simulation). By using a CBCT data with a 0.075 mm voxel size, the image reconstruction software produces an extremely fine mesh with a minimum of 0.075-0.08 mm element size. However, after the reconstruction of all anatomical components, 3D models assembling and the surface smoothing, the global element size reaches 0.08-0.116 mm (confirmed by the finite elements analysis software during the mesh verify phase). A large number of elements and nodes usually associates with a reduced global elements size and vice-versa (e.g., Fields’s et al. [15], 23565-32812 elements, 1.2 mm mesh size vs. 5.06-6.05 million C3D4 elements, 0.08-0.116 mm element size for herein). Anatomical irregular shape of the 3D model components imposed a discretization using of tetrahedral finite elements. In ABAQUS library are available both C3D4 (first order linear interpolation tetrahedron element) and C3D10 (second order quadratic interpolation tetrahedron element). It is acknowledged that C3D4 elements show additional rigidity, nonetheless, can be sur-passed by a fine discretization of the model. However, the use of C3D10 elements despite their better accuracy imply an extremely high compute power. Moreover, the anatomical irregular shape makes impossible the type of stress/displacement continuum shell (SC6R - six node triangular in plane continuum shell wedge) discretization while their use is recommended in bending dominated problems.”

Round 2

Reviewer 1 Report

All issues solved.